# Pathological and Molecular Features of Nodal Peripheral T-Cell Lymphomas

**DOI:** 10.3390/diagnostics12082001

**Published:** 2022-08-18

**Authors:** Akira Satou, Taishi Takahara, Toyonori Tsuzuki

**Affiliations:** Department of Surgical Pathology, Aichi Medical University Hospital, Nagakute 480-1195, Japan

**Keywords:** nodal PTCL, ALCL, AITL, nodal PTCL-TFH, ATLL, PTCL-NOS, genetics, pathology

## Abstract

Peripheral T-cell lymphomas (PTCLs) are uncommon neoplasms derived from mature T cells or NK cells. PTCLs comprise numerous disease entities, with over 30 distinct entities listed in the latest WHO classification. They predominantly affect adults and elderly people and usually exhibit an aggressive clinical course with poor prognosis. According to their presentation, PTCLs can be divided into nodal, extranodal or cutaneous, and leukemic types. The most frequent primary sites of PTCLs are lymph nodes, with over half of cases showing nodal presentation. Nodal PTCLs include ALK-positive and ALK-negative anaplastic large cell lymphoma; nodal T-cell lymphoma with T follicular helper cell origin; and PTCL, not otherwise specified. Adult T-cell leukemia/lymphoma also frequently affects lymph nodes. Recent pathological and molecular findings in nodal PTCLs have profoundly advanced the identification of tumor signatures and the refinement of the classification. Therefore, the therapies and pathological diagnosis of nodal PTCLs are continually evolving. This paper aims to provide a summary and update of the pathological and molecular features of nodal PTCLs, which will be helpful for diagnostic practice.

## 1. Introduction

Peripheral T-cell lymphomas (PTCLs) are uncommon neoplasms derived from mature T cells or NK cells. PTCLs collectively account for 10–15% of all non-Hodgkin lymphomas and usually exhibit an aggressive clinical course with poor prognosis [1,2,3]. They predominantly affect adults and elderly people and are more common in Asian than in Western countries.

PTCLs comprise numerous disease entities, with over 30 distinct entities of T-cell or NK-cell origin listed in the latest World Health Organization (WHO) classification of Tumors of Haematopoietic and Lymphoid Tissues [1]. In addition to histological and immunohistochemical signatures, recent molecular studies have profoundly contributed to the identification of tumor signatures and the refinement of the classification. According to their presentation, PTCLs can be divided into nodal, extranodal or cutaneous, and leukemic types [1,4]. The biological features of PTCLs are somewhat associated with the site of primary presentation. The most frequent primary sites of PTCLs are lymph nodes, with over half of cases showing nodal presentation.

In the current review, we summarize the pathological and molecular features of PTCLs that mainly or frequently show nodal presentation. Nodal PTCLs include ALK-positive (ALK^+^) and ALK-negative (ALK^−^) anaplastic large cell lymphoma (ALCL); nodal T-cell lymphoma with T follicular helper (TFH) cell origin; and PTCL, not otherwise specified (PTCL-NOS). Adult T-cell leukemia/lymphoma (ATLL) also frequently affects lymph nodes.

## 2. ALCL

According to the WHO classification, ALCLs are classified into four entities based on their clinical presentation and ALK expression. All four entities share the pathological features of strong and diffuse CD30 expression and the presence of characteristic cells called “hallmark cells” [1]. Systemic ALCL includes ALK^+^ ALCL and ALK^−^ ALCL, while localized ALCL includes primary cutaneous (pc) ALCL and breast implant-associated (BIA) ALCL. ALK^+^ ALCL expresses the ALK protein due to chromosomal translocations involving the ALK gene. ALK^−^ ALCL is systemic ALCL without ALK expression. pcALCL is a skin-limited disease and a pcCD30^+^ T-cell lymphoproliferative disorder. About 10% of patients with pcALCL exhibit extracutaneous lesions that mainly affect the regional lymph nodes [5,6]. BIA ALCL is a new entity that is now included as a provisional entity in the WHO classification. This disease arises in association with a breast implant and is usually limited to a seroma cavity around the implant and pericapsular fibrous tissue [7]. Among these four entities, ALK^+^ and ALK^−^ ALCL frequently appear as nodal lesions at primary presentation. Next, we further describe ALK^+^ and ALK^−^ ALCL.

### 2.1. ALK^+^ ALCL

#### 2.1.1. Clinical Features

ALK^+^ ALCL mainly occurs in children and young adults and shows a male-to-female ratio of 3:2. Patients frequently have B-symptoms, especially high fever, and present with stage III-IV disease. Most patients present with nodal lesions, although extranodal involvement is often observed (60%). Overall, ALK^+^ ALCL has a better prognosis than ALK^−^ ALCL, with a long-term overall survival of 80% [8,9].

#### 2.1.2. Morphological and Immunohistochemical Features

ALK^+^ ALCLs are characterized by a broad spectrum of morphological and cytological features. However, all cases include a variable proportion of hallmark cells, which are large-sized cells with eccentric kidney- or horseshoe-shaped nuclei. They often have a paranuclear eosinophilic region [8,10]. According to the WHO classification, ALK^+^ ALCLs include five morphological patterns: common (60%), lymphohistiocytic (10%), small-cell (5–10%), Hodgkin-like (3%), and composite (15%) patterns [1].

The common pattern shows a sheet-like and cohesive growth pattern of large tumor cells with irregular nuclei, which may resemble Hodgkin–Reed–Sternberg cells. Some of them have hallmark cell features (Figure 1a), and the large neoplastic cells have rich cytoplasm with clear, basophilic, or eosinophilic features. Tumor cell sinusoidal involvement is often observed in lymph node lesions, which may mimic a metastatic malignant tumor.

In the lymphohistiocytic pattern, neoplastic cells are scattered and admixed with abundant reactive histiocytes and small lymphocytes (Figure 1b) [11,12]. These reactive components may mask the tumor cells, leading to a misdiagnosis of a reactive lesion. Compared to the common pattern, this variant usually includes smaller tumor cells that often cluster around blood vessels. The small cell pattern includes numerous small-to-medium-sized tumor cells with irregular nuclei, pale cytoplasm, and a distinct membrane borderline. There are few hallmark cells, which tend to cluster around blood vessels [13]. These lymphohistiocytic and small cell patterns are closely related and often admixed [11,14]. Among the ALK^+^ ALCLs, these two morphological variants are associated with an unfavorable prognosis and leukemic presentation [11].

The Hodgkin-like pattern histologically resembles nodular sclerosis Hodgkin lymphoma [15]. Other rare patterns include cases with tumor cells showing giant-cell, sarcomatoid, or spindle-cell features [16,17]. The composite pattern includes the presence of two or more patterns in one lymph node.

Immunohistochemically, in all ALK^+^ ALCLs, the tumor cells are positive for CD30 on the cell membrane and the Golgi region (Figure 1c) and express ALK (Figure 1d). Large tumor cells show strong CD30 expression, while smaller tumor cells may show only weak or no CD30 expression. Therefore, the common pattern, which predominantly comprises large tumor cells, exhibits diffuse and strong CD30 expression (Figure 1c). In contrast, the lymphohistiocytic and small cell patterns may exhibit heterogenous CD30 expression (Figure 1e). As CD30 and ALK are definitional markers for ALK^+^ ALCL diagnosis, their evaluation is useful for accurate diagnosis, particularly to highlight the tumor cells of morphologic variants, in which hallmark cells may be scarce or masked by abundant inflammatory infiltrates (Figure 1f). Most cases are positive for EMA. Although most ALK^+^ ALCLs show a monoclonal rearrangement of the TCR genes, the tumor cells often lack T-cell antigen expression [18]. The majority of cases are negative for CD3, and many are also negative for CD5 and CD7. CD2 and CD4 are more useful and are positive in a substantial proportion of cases. Some cases may have an apparent null-cell immunophenotype. Most cases, even of the null-cell type, express cytotoxic markers, such as TIA-1, granzyme B, and/or perforin. CD8 is usually negative. By definition, ALCLs are negative for EBV.

#### 2.1.3. Molecular Features

Chromosomal translocations involving the *ALK* gene on 2p23 lead to the expression of an ALK fusion protein. Approximately 80% of ALK^+^ ALCLs harbor t(2;5)(p23;q35), which results in the fusion of *ALK* and *NPM1*, and less common mutations include t(1;2)(q25;p23)/*TPM3-ALK*, inv(2)(p23q35)/*ATIC-ALK*, and other rare variants [1,19]. The ALK staining pattern differs according to the variant of translocation (Table 1). The aberrant expression of ALK fusion proteins leads to constitutive ALK tyrosine kinase activity and downstream activation of the JAK-STAT3 pathway [20].

### 2.2. ALK^−^ ALCL

#### 2.2.1. DUSP22-Rearranged, TP63-Rearranged, and Triple-Negative ALCL

All systemic ALCLs lacking ALK expression are categorized as ALK^−^ ALCL. Therefore, ALK^−^ ALCL is a heterogenous entity. Next-generation sequencing has revealed two recurrent rearrangements in ALK^−^ ALCL: *DUSP22* and *TP63* rearrangement [21]. The investigation of a large cohort revealed *DUSP22* and *TP63* rearrangements, respectively, in 30% and 8% of ALK^−^ ALCLs [22]. These two rearrangements are associated with significant clinical features, as described below. Based on these findings, ALCLs are divided into four genetic subtypes: *ALK*-rearranged, *DUSP22*-rearranged, *TP63*-rearranged, and triple-negative ALCL. These rearrangements are mutually exclusive, with the exception of a rare case harboring both *DUSP22* and *TP63* rearrangements [23].

#### 2.2.2. Clinical Features

ALK^−^ ALCL generally occurs in older adults, with a male-to-female ratio of 3:2. Most patients present with stage III-IV disease and B symptoms [24]. Compared to ALK^+^ ALCL, ALK^−^ ALCL less frequently exhibits extranodal involvement. Overall, ALK^−^ ALCL has a worse prognosis than ALK^+^ ALCL but a better prognosis than PTCL-NOS [24]. Notably, *DUSP22*-reararaged ALCLs have a favorable prognosis, with a 5-year overall survival rate of 90%, which is comparable to that of ALK^+^ ALCL. In contrast, *TP63*-rearranged cases have a dismal prognosis, with a 5-year overall survival rate of 17% [22,25].

#### 2.2.3. Morphological and Immunohistochemical Features

Morphologically, ALK^−^ ALCL is similar to the common pattern of ALK^+^ ALCL, with the presence of hallmark cells, and frequently exhibits a cohesive sheet of large tumor cells with irregular nuclei and sinusoidal involvement. ALK^−^ ALCL is not categorized into different morphological variants [1,8]. Compared to other ALK^−^ ALCLs, *DUSP22*-rearranged cases are more likely to exhibit a sheet-like growth pattern and doughnut-cell appearance, and they are less likely to display pleomorphism [26].

Immunohistochemically, the tumor cells are strongly and diffusely positive for CD30 on the cell membrane and Golgi region. By definition, ALK is not expressed. Similar to ALK^+^ ALCL, ALK^−^ ALCL frequently shows the loss of T-cell antigens. However, the majority of ALK^−^ ALCLs express one or more T-cell markers and exhibit some suggestion of a T-cell phenotype. Many cases express cytotoxic markers and EMA but do so less frequently than ALK^+^ ALCL [24]. *DUSP22*-rearranged cases show distinct immunohistochemical features; they are generally negative for cytotoxic markers and EMA, while they are generally positive for CD2 and CD3 [22], and they generally express LEF1 [27]. *TP63*-rearranged cases always show p63 expression, but this is not specific to them [28]. Therefore, p63 cannot be used as a diagnostic marker of *TP63-* rearranged cases but can be used as a screening tool.

#### 2.2.4. Molecular Features

ALK^−^ ALCLs frequently exhibit JAK-STAT3 pathway activation, which is also a common feature of ALK^+^ ALCL. Recurrent mutations of *JAK1* and *STAT3* have been identified in some cases of ALK^−^ ALCL [29]. A subset of cases also show rearrangements of non-ALK tyrosine kinase genes (*TYK2, ROS1,* and *FRK*), which result in JAK-STAT3 activation [30]. Notably, *DUSP22*-rearranged ALCLs exhibit distinct molecular features compared to other ALCLs. They lack STAT3 activation and are characterized by hypomethylation and an immunogenic phenotype that includes the overexpression of immunogenic cancer testis antigen, CD58, and HLA class II and the minimal expression of PD-L1 [29]. A hotspot *MSC*
^E116K^ mutation occurs almost exclusively in *DUSP22-*rearranged ALCLs [31] (Table 2).

## 3. Nodal T-Cell Lymphoma with TFH Cell Origin

According to the WHO classification, nodal T-cell lymphomas with TFH cell origin include three lymphoma types: AITL, nodal PTCL with a TFH phenotype (nPTCL-TFH), and follicular T-cell lymphoma (FTCL) [1]. These lymphomas exhibit a TFH phenotype and are considered a neoplastic counterpart of TFH cells. According to the WHO classification, a TFH lineage of neoplastic cells is indicated by the expression of at least two—and, ideally, three or more—TFH markers.

### 3.1. AITL

#### 3.1.1. Clinical Features

AITL is a relatively common type of T/NK-cell lymphoma, accounting for 25–30% of such cases [32,33]. AITL most frequently occurs in middle-aged and elderly individuals and presents as a systemic disease. Patients present with generalized lymphadenopathy, and frequently involved extranodal sites include the spleen, liver, skin, and bone marrow. Patients may also exhibit advanced-stage disease, hepatosplenomegaly, and systemic symptoms, such as fever, skin rash, weight loss, and arthritis. Laboratory tests often show immune abnormalities, including polyclonal hypergammaglobulinemia, elevated rheumatoid factor, Coombs-positive hematolytic anemia, positive anti-smooth muscle antibodies, and others [34,35,36]. The prognosis is generally poor, with a median survival of <3 years [33,35,36].

#### 3.1.2. Morphological and Immunohistochemical Features

In AITLs, the lymph nodes are characterized by partial or complete architectural effacement and the marked proliferation of high endothelial venules (HEVs) (Figure 2a). The neoplastic cells of AITLs are typically intermediate-to-medium-sized, with mild nuclear atypia and moderately abundant clear-to-pale cytoplasm (Figure 2b). Tumor cells are accompanied by abundant polymorphous infiltrate, including reactive small lymphocytes, immunoblasts, plasma cells, histiocytes, and eosinophils. The tumor cells may show low cellularity and may be obscured by reactive components. The neoplastic cells tend to form clusters in the vicinity of HEVs. AITLs are often associated with the expansion of extrafollicular follicular dendritic cell (FDC) meshworks, which can be highlighted by immunohistochemistry with CD21 or CD23. These FDC meshworks are most prominent around the HEVs.

Three histological patterns are recognized: patterns 1, 2, and 3 [37]. Pattern 1 is histologically characterized by many hyperplastic follicles surrounded by atypical lymphocytes (Figure 2c). The interfollicular area exhibits HEV proliferation and nuclear cells with a polymorphous inflammatory background. This pattern is difficult to distinguish from reactive follicular hyperplasia. In pattern 2, the follicles are depleted, and regressive follicles remain (Figure 2d). Neoplastic cell expansion is easily recognized in pattern 2 compared to pattern 1. Pattern 3 is the most common pattern and is characterized by a totally or mostly effaced lymph node architecture, with regressed follicles potentially remaining in the periphery of the lymph nodes (Figure 2e).

Immunohistochemically, the neoplastic cells are positive for pan-T-cell markers, such as CD2 and CD3. Most cases are positive for CD4 (Figure 3a) and negative for CD8 (Figure 3b), which is consistent with the TFH origin of AITL. Expressions of pan-T-cell antigens, particularly CD5 or CD7, are frequently downregulated or absent. The tumor cells show positive staining for several TFH markers, including PD-1, ICOS, CXCL13, CD10, and BCL6 (Figure 3c–e). The TFH markers exhibit variable sensitivity and specificity; PD1 and ICOS are more sensitive but less sensitive, while CXCL13 and CD10 are more specific but less sensitive [38,39].

Immunoblasts in the inflammatory background of AITLs are non-neoplastic B cells and may be positive for EBV (Figure 3f). Previous studies demonstrate that EBV^+^ B cells are present in 65–95% of AITL cases [35,40,41,42]. The number of EBV^+^ B cells varies among cases.

#### 3.1.3. Molecular Features

Clonal rearrangements of TCR are found in 75–90% of AITL cases. Additionally, *IgH* rearrangements are detected in approximately 25–30% of AITL cases. Recent studies show that AITLs have recurrent mutations in genes involved in epigenetic pathways, including *TET2* (40–80%), *IDH2^R172^* (30%), and *DNMT3A* (30%) [43,44,45,46,47]. Other studies have reported hotspot *RHOA^G17V^* mutations and mutations in genes involving the TCR signaling pathway, such as *PLCG1*, *CD28*, *FYN*, and *VAV1* [44,48,49,50]. Notably, *TET2*, *DNMT3A,* and *IDH2^R172^* mutations often co-occur in the same case of AITL. In one study, all AITL cases with *RHOA^G17V^* mutations also had *TET2* mutation [43]. Furthermore, *TET2* and *DNMT3A* mutations have been identified in the non-tumor cells of patients with AITL and even in the blood cells of healthy individuals [51,52]. On the other hand, the *IDH2^R172^* and *RHOA^G17V^* mutations are confined to tumor cells. Based on these findings, a multistep model of tumor development in AITL has been proposed [53].

AITLs with *IDH2^R172^* or *RHOA^G17V^* mutations are associated with several morphological and immunohistochemical features. AITL cases with *IDH2^R172^* mutations are characterized by prominent medium-to-large clear cells and a strong expression of TFH markers, particularly CD10 and CXCL13 [54]. AITL cases with *RHOA^G17V^* mutations are characterized by a higher density of HEVs, greater FDC expansion, and a more pronounced TFH phenotype compared to wild-type cases [55,56].

### 3.2. FTCL and nPTCL-TFH

#### 3.2.1. Clinical Features

FTCL is a rare form of PTCL with clinical features similar to those of AITL [57,58]. FTCL occurs in middle-aged and elderly individuals. Patients with FTCL present with generalized lymphadenopathy, splenomegaly, B symptoms, skin rash, and advanced-stage disease. The prognosis appears to be poor. Laboratory tests sometimes reveal immune abnormalities. nPTCL-TFH is not as rare as FTCL. A subset of cases were initially diagnosed as PTCL-NOS and were then identified as having a TFH phenotype.

#### 3.2.2. Morphological and Immunohistochemical Features

In FTCL, the lymph nodes show a nodular/follicular growth pattern (Figure 4a). FTCLs are divided into two histologic patterns: follicular lymphoma-like (FL-like) and the progressive transformation of germinal center-like (PTGC-like) [57,59,60]. In the FL-like pattern, aggregations of intermediate-sized neoplastic cells with round nuclei and pale cytoplasm are observed inside the follicles (Figure 4b), which lack the normal features of follicular B cells. In the PTGC-like pattern, medium-sized neoplastic cells with pale cytoplasm form aggregates surrounded by IgD^+^ small B cells in an expanded mantle zone. Unlike AITLs, FTCLs lack a polymorphic inflammatory background and HEV proliferation. B immunoblasts are often observed, some of which morphologically and immunohistochemically mimic HRS cells [61]. Immunostaining for CD21 or CD23 can highlight the FDC meshworks in nodular/follicular structures (Figure 4c).

nPTCL-TFH includes cases with a TFH phenotype but without sufficient pathological features to be diagnosed as AITL. Therefore, although some AITL-like features may be recognized, nPTCL-TFH usually exhibits the diffuse proliferation of neoplastic cells without an abundant inflammatory background, HEV proliferation, or FDC expansion.

#### 3.2.3. Molecular Features

The mutational profiles of FTCL and nPTCL-TFH overlap with AITL. Mutations in *TET2*, *DNMT3A*, and *RHOA^G17V^* are also detected in FTCL and nPTCL-TFH [49,62,63,64]. *TET2* mutations seem to be more frequent in FTCL and nPTCL-TFH. *IDH2^R172^* mutations are restricted to FTCLs and AITLs. About 20% of FTCL cases carry a characteristic chromosomal translocation t(5;9)(q33;q22), which results in *ITK-SYK* fusion [65]. Outside of FTCL, the *ITK-SYK* fusion has only rarely been detected in AITL (Table 3).

### 3.3. TFH Lymphomas with Hodgkin–Reed–Sternberg (HRS)-Like Cells

Immunoblasts that mimic HRS cells occasionally appear in the background of TFH lymphomas (AITL, nPTCL-TFH, and FTCL) (Figure 5a,b) [61,66,67,68]. Therefore, pathologists may easily misdiagnose THF lymphomas with HRS-like cells as classic Hodgkin lymphoma (cHL). Furthermore, the standard immunostaining used for cHL diagnosis cannot differentiate these HRS-like cells from true HRS cells. We recently revealed that PD-L1 expression is highly frequent in HRS cells (Figure 5c) and rare in HRS-like cells (Figure 5d) [69]. Based on these findings, it was suggested that PD-L1 immunohistochemistry is useful for differentiating these two diseases.

## 4. ATLL

ATLL is a mature T-cell neoplasm caused by human T-cell leukemia virus type 1 (HTLV-1) retrovirus [1]. Therefore, ATLL frequency is linked to the prevalence of HTLV-1 carriers. ATLL is endemic in several parts of the world, particularly southwest Japan, the Caribbean, Central and South America, and central Africa. Carriers are usually infected by HTLV-1 early in life. ATLL occurs after a long latency, and the cumulative incidence is estimated to be 2–5% among HTLV-1 carriers [70,71,72]. HTLV-1 infection alone is not sufficient for ATLL development, and TAX and HBZ play key roles in ATLL pathogenesis [73]. TAX is a viral oncoprotein that is needed for ATLL initiation and early proliferation, while HBZ is required for neoplasm maintenance. Based on clinical behavior, ATLL is subclassified into four clinical variants: acute, lymphomatous, chronic, and smoldering [74]. Nodal involvement occurs in the acute and lymphomatous variants.

### 4.1. Clinical Features

Due to the long latency period, ATLL occurs only in adults, with a median onset age of 68 years (ranging from around 30–90 years) [75]. The acute and lymphomatous variants are regarded as aggressive types [74]. Patients with aggressive types of ATLL often exhibit cutaneous lesions, hepatosplenomegaly, and generalized lymphadenopathy. Other frequently involved sites include the central nervous system, gastrointestinal tract, bone, and lung. Laboratory tests reveal elevations of LDH and sIL-2R, hypercalcemia, leukocytosis, and eosinophilia. Due to the associated T-cell immunodeficiency, many patients suffer from frequent opportunistic infections, such as *Pneumocystis jirovecii* pneumonia, disseminated herpes zoster, and strongyloidiasis. Aggressive ATLLs have a dismal prognosis, with a median survival time of less than one year [76].

The chronic and smoldering variants are regarded as indolent types. Compared to the aggressive variants, they are characterized by a more indolent clinical course and a better prognosis. However, about 25% of indolent ATLLs progress to an acute phase [76,77].

### 4.2. Morphological and Immunohistochemical Features

The lymph node lesions of ATLLs exhibit a broad spectrum of cytological and morphological features. The lymph nodes are characterized by partial or complete architectural effacement and the diffuse proliferation of neoplastic cells. Tumor cells are usually medium-to-large and pleomorphic (Figure 6a), and small-sized tumor cells with pleomorphic nuclei may also be present. Additionally, anaplastic-like (Figure 6b), multilobed giant cell-like, and blast-like tumor cells are occasionally observed. Rare cases may exhibit AITL-like histological features [78]. Immunohistochemically, neoplastic cells of ATLL express pan-T-cell antigens (CD2, CD3, and CD5) but usually lack CD7 expression (Figure 6c). Most cases are positive for CD4 (Figure 6d) and negative for CD8 (Figure 6e). The majority of ATLLs are positive for CCR4, which is a target of the humanized monoclonal antibody mogamulizumab [79]. About 40% of cases show positive staining for FOXP3, a regulatory T-cell marker [80,81].

Other rare morphological variants include Hodgkin-like ATLL [82,83] and ATLL with HTLV-1-infected HRS-like cells (ATLL-HH) [84]. Hodgkin-like ATLL is considered an incipient or early neoplastic phase of ATLL. In Hodgkin-like ATLL, the lymph nodes exhibit small-to-medium-sized lymphocytes with mild nuclear atypia, which diffusely proliferate in expanded paracortical areas (Figure 7a). These atypical lymphocytes are positive for CD3 and CD4 and negative for CD8. Scattered HRS-like cells are observed throughout the paracortical areas (Figure 7a). These HRS-like cells are of the B-cell lineage (Figure 7b), show positive staining for CD30 (Figure 7c) and/or CD15 (Figure 7d), and are frequently infected by EBV (Figure 7e). Notably, HTLV-1 infection is limited to CD4^+^ T cells, and ultrasensitive RNA in situ hybridization for HBZ shows that HRS-like cells are not infected by HTLV-1 (Figure 7f), implying that HRS-like cells are not neoplastic cells.

ATLL-HH cases are morphologically similar to CHL, exhibiting scattered HRS-like cells intermingled with abundant inflammatory cells (Figure 8a) [84]. In contrast to Hodgkin-like ATLL, the HRS-like cells of ATLL-HH are infected with HTLV-1, as demonstrated by ultrasensitive RNA in situ hybridization for *HBZ* (Figure 8b). Therefore, these HRS-like cells of ATLL-HH are considered neoplastic cells. Immunohistochemically, these HRS-like cells show positive staining for CD30 (Figure 8c), CD15 (Figure 8d), CD25, and MUM1 and are negative for B-cell markers (Figure 8e), pan-T-cell markers, and EBV. They are frequently positive for CD4 and fascin (Figure 8f).

### 4.3. Molecular and Microenvironmental Features

Recent studies have revealed the hallmark genetic features of ATLL, including genetic alterations involved in the TCR/NF-κβ pathway and the immune escape mechanism. Up to 90% of ATLL cases harbor genomic alterations affecting the TCR/NF-κβ pathway [85,86,87,88]. These include genetic alterations of a proximal component of TCR signaling (e.g., *FYN*, *PLCG1*, and *VAV1*), of the NF-κβ pathway (e.g., *PRKCB* and *CARD11*), and of downstream signaling (e.g., *IRF4* and *RHOA*). Around 10% of ATLLs exhibit in-frame fusions related to CD28 (*CTLA4-CD28* and *ICOS-CD28*) [89], which contribute to NF-κβ pathway activation via CD28 costimulatory signaling.

Immune escape mechanisms also play key roles in ATLL pathogenesis. Over half of ATLL cases exhibit alterations in genes associated with immune response modulation. Copy number gains of *PD-L1* are frequently detected [85], and the disruption of the 3′-UTR in *PD-L1* has been found in 27% of ATLL cases [90]. These alterations of *PD-L1* lead to the overexpression of PD-L1 on tumor cells and promote tumor development by inactivating tumor-specific CD8^+^ T cells via the PD-1/PD-L1 axis. Other genes related to immune response modulation are also altered—including deletions and mutations in *HLA-A*, *HLA-B*, and *B2M*, which are associated with MHC class I, and alterations in *CD58* and *FAS*, which are associated with T-cell and NK-cell immune responses [85,91]. In a large ATLL cohort, immunohistochemical analysis revealed that 7.4% of cases expressed PD-L1 on tumor cells [92]. This neoplastic PD-L1^+^ group had a poor prognosis compared to the neoplastic PD-L1^−^ group. Within the neoplastic PD-L1^−^ group, the prognosis was better for patients exhibiting microenvironmental PD-L1 expression compared to those lacking microenvironmental PD-L1 expression. In another immunohistochemical study, the prognosis was better among ATLL cases with HLA class II expression on tumor cells than among those without this expression [93]. Furthermore, cases with both neoplastic HLA class II expression on tumor cells and microenvironmental PD-L1 positivity have a significantly better prognosis than the other groups.

*CCR4* mutations have been identified in about 30% of ATLL cases [85,94,95]. These mutations result in the up-regulation of CCR4 expression and the activation of the PI3K/AKT pathway. Immunohistochemical analyses show CCR4 expression in 90% of ATLL cases, and CCR4^+^ patients are likely to have a poorer prognosis than CCR4^−^ patients [79]. ATLL patients with CCR4 expression are good candidates for anti-CCR4 antibody (mogamulizumab) therapy, which shows some effects in ATLL [96].

Clinicogenetic studies of ATLL have highlighted the correlations between genetic alterations and clinical features [97,98]. Mutations of *IRF4* and *TP53*, the amplification of *PD-L1*, and deletions of *CDKN2A* are more frequently detected in aggressive variants compared to indolent variants. In aggressive variants, *PD-L1* amplifications and *PDKCB* mutation are independent adverse prognostic factors, according to multivariate analysis. Among indolent variants, *IRF4* mutations, *PD-L1* amplifications, and *CDKN2A* deletions are significantly associated with a poor outcome. A recent report highlights the geographical heterogeneity between ATLL cases in North America and Japan [86]. ATLL cases in North America exhibited more frequent mutations related to epigenome and histone modification. In particular, mutations of EP300 were detected in 20% of ATLL cases in North America, compared to 6% in Japanese cohorts (Table 4).

## 5. PTCL-NOS

PTCL-NOS is heterogenous category of nodal and extranodal T-cell lymphomas that do not fulfill the criteria for any specific PTCL entity. The differentiation between nPTCL-TFH and PTCL-NOS requires the immunohistochemical analysis of TFH markers.

### 5.1. Clinical Features

PTCL-NOS is the most common entity among PTCLs, accounting for approximately 30% of PTCLs. Most cases occur in adults, and children are rarely affected by this disease [1]. The presentation usually includes nodal manifestation, with frequent extranodal involvement. The tumor may primarily occur in extranodal sites, with the most frequently involved sites being the skin and gastrointestinal tract. Most patients present with stage III–IV disease and B symptoms. Eosinophilia, pruritus, or hemophagocytic syndrome may appear as paraneoplastic symptoms. The prognosis of PTCL-NOS is generally poor, with a 5-year OS rate of 20–30% [4,33,99].

### 5.2. Morphological and Immunohistochemical Features

Since PTCL-NOS is a heterogenous category, the morphological and immunohistochemical features are also heterogenous. Nodal PTCL-NOS usually exhibits diffuse architectural effacement, with some remnants of B cells in the periphery of lymph nodes. Some cases may exhibit an interfollicular or paracortical growth pattern. The cytological spectrum is extremely broad. Pleomorphic medium- and large-sized lymphocytes with irregular nuclei are most frequently observed (Figure 9a). Less common observations include the predominant proliferation of small-sized atypical lymphocytes with irregular nuclei (Figure 9b) or of large cells with immunoblast-like features (Figure 9c). PTCL-NOS is often accompanied by reactive components, including small lymphocytes, histiocytes, eosinophils, B cells, and plasma cells. Scattered EBV-infected B blasts or HRS-like cells may be present.

Immunohistochemically, the tumor cells are usually positive for pan-T-cell markers (CD2, CD3, CD5, and CD7), although one or several (particularly, CD5 or CD7) may be downregulated or absent. The majority of cases are CD4^+^/CD8^−^, while CD4^−^/CD8^+^ cases are less frequent, and CD4^−^/CD8^−^ or CD4/CD8^+^ cases have also been described.

A lymphoepithelioid variant of PTCL-NOS, also known as Lennert lymphoma, is histologically characterized by numerous clusters of epithelioid histiocytes and the infiltration of small-to-medium-sized lymphocytes with slight nuclear irregularities (Figure 10a). This tumor usually exhibits a cytotoxic phenotype (Figure 10b). Approximately 70–80% of cases are positive for cytotoxic molecules [99,100,101], and 50–80% show CD8 positivity (Figure 10c). This variant exhibits a better prognosis than other PTCL-NOS variants [99,102].

Nodal cytotoxic molecule (CM)-positive PTCL-NOS is characterized by the expression of at least one of CM: TIA-1, granzyme B, or perforin. Previous studies have revealed a cytotoxic phenotype in 15–40% of PTCL-NOS cases [100,103,104]. Compared to CM-negative cases, nodal CM-positive PTCL-NOSs are characterized by a younger onset, a poorer performance status, more frequent B symptoms, higher serum LDH, more frequent extranodal involvement, and more frequent EBV positivity. CM-positive cases have a worse prognosis than CM-negative cases [104]. One recent study reported that, among nodal EBV^−^ CM-positive PTCL-NOS cases, the CD5^+^ cases showed an indolent clinical course [105].

As mentioned above, nodal CM-positive PTCL-NOS includes EBV^+^ cases. Such cases are now considered a distinct entity: nodal EBV^+^ cytotoxic T-cell lymphoma (CTL) [106,107,108], which accounts for 21% of PTCL-NOS [104]. Morphologically, these cases all show high-grade morphology, and half exhibit a centroblast-like appearance. Necrosis and an angiocentric pattern are infrequent, in contrast to extranodal NK/T cell lymphoma. Immunohistochemically, they are usually CD8^+^/CD56^−^, and the majority are TCRαβ type, although a subset are TCRγδ type or TCR-silent type [107]. In general, nodal EBV^+^ CTL is characterized by an aggressive clinical course.

### 5.3. Molecular and Microenvironmental Features

A gene expression study identified two molecular subgroups in PTCL-NOS: the GATA3 subgroup, characterized by the high expression of *GATA3* and its target genes (*CCR4*, *IL18RA*, *CXCR7*, and *IK*); and the TBX21 subgroup, characterized by the high expression of *TBX21* and its target genes (*CXCR3*, *IL2RB*, *CCL3*, and *IFNγ*) [109]. Notably, the GATA3 subgroup exhibited a worse prognosis than the TBX21 subgroup. Within the TBX21 subgroup, cases with a cytotoxic gene signature had poor clinical outcomes. In a subsequent study, genomic copy number analysis and target sequencing revealed that the GATA3 subgroup is characterized by higher genomic complexity and frequent losses or mutations of tumor suppressor genes, such as *CDKN2A*, *TP53*, and *PTEN* [110]. The GATA3 subgroup is also characterized by frequent gains or amplifications of *STAT3* and *MYC*. An immunohistochemistry (IHC) algorithm was generated, in which four antibodies (GATA3, TBX21, CCR4, and CXCR3) are used to identify these two subgroups [111].

A recent comprehensive molecular study identified a subgroup of PTCL-NOS with *TP53* and/or *CDKN2A* mutations and deletions [112]. This subgroup is characterized by a poor prognosis, extensive chromosomal instability, and mutations in genes associated with immune escape and transcriptional regulation (e.g., *HLA-A*, *HLA-B*, *CD58*, and *IKZF2*). In another study, whole genome sequencing and FISH analysis revealed that the most frequent aberrations among PTCL-NOS cases were *CDKN2A* deletions (46%) and *PTEN* deletions (26%) [113]. PTCL-NOS cases with these aberrations are also associated with poor prognosis. This study also demonstrated that the co-occurrence of *CDKN2A* and *PTEN* deletions was exclusively detected in PTCL-NOS and that these aberrations were rarely detected in AITL and ALCL. Previous studies of PTCL-NOS have reported recurrent mutations of genes involved in epigenetic regulation, such as *TET2* and *DNMT3A*; however, these studies have included nPTCL-TFH cases. Recent studies that have excluded nPTCL-TFH cases highlight that *DNMT3A* mutations are rare in PTCL-NOS, while *TET2* mutations occur in around 20% of cases [45,112,113]. Similar to other types of T-cell lymphomas, PTCL-NOS can reportedly harbor mutations in genes involved in the TCR signaling pathway—such as *PLCG1*, *CD28*, and *VAV1*—which are associated with aberrations of *TP53* and *CDKN2A* [48,112]. In another recent study, target capture sequencing identified recurrent mutations of FAT1 in 39% of PTCL-NOS cases [114]. *FAT1* is a tumor suppressor gene encoding a member of the cadherin superfamily, which binds to β-catenin and inhibits nuclear localization. Patients with *FAT1* mutation showed inferior outcomes compared to patients with wild-type *FAT1*.

In a study focused on microenvironment, the use of the nCounter system revealed a favorable clinical outcome among patients with PTCL-NOS who have both B-cell and dendritic cell (DC) signatures (BD subgroup), compared to an extremely poor prognosis among patients who have neither B-cell nor DC-cell signatures (non-BD subgroup) [115]. Half of the patients in the non-BD subgroup exhibited a macrophage signature (Table 5).

## 6. Conclusions

Recent pathological and molecular findings in nodal PTCLs have profoundly advanced the identification of tumor signatures and the refinement of the classification. Therefore, the therapies and pathological diagnosis of nodal PTCLs are continually evolving. In this paper, we provide a summary and update of the pathological and molecular features of nodal PTCLs, which will be helpful for diagnostic practice.

## Figures and Tables

**Figure 1 diagnostics-12-02001-f001:**
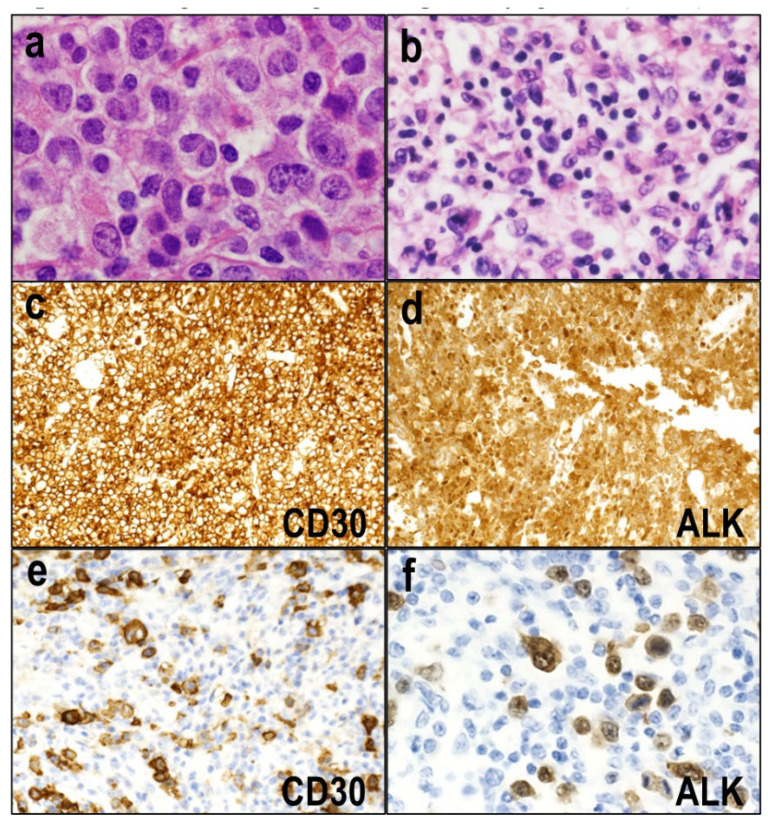
ALK-positive anaplastic large cell lymphoma (ALCL). (**a**) The common pattern shows a sheet-like and cohesive growth pattern of large tumor cells with irregular nuclei, some of which have hallmark cell features (HE × 400). (**b**) In the lymphohistiocytic pattern, neoplastic cells are scattered and admixed with abundant reactive histiocytes and small lymphocytes (HE × 400). (**c**) The common pattern shows diffuse and strong CD30 expression (×200). (**d**) In all ALK^+^ ALCL cases, the tumor cells express ALK (×200). (**e**) The lymphohistiocytic and small-cell patterns may show heterogenous CD30 expression patterns (×200). (**f**) ALK immunohistochemistry highlights the tumor cells of morphologic variants, which may contain scarce hallmark cells.

**Figure 2 diagnostics-12-02001-f002:**
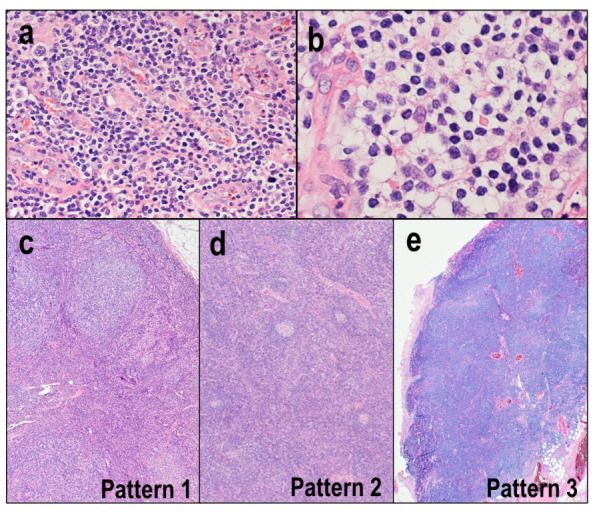
Angioimmunoblastic T-cell lymphoma. (**a**) Lymph nodes are characterized by partial or complete architectural effacement and a marked proliferation of high endothelial venules (HE × 200). (**b**) Tumor cells are typically intermediate-to-medium-sized, with mild nuclear atypia and moderately abundant clear-to-pale cytoplasm (HE × 400). (**c**) Pattern 1 is histologically characterized by many hyperplastic follicles surrounded by atypical lymphocytes (HE × 100). (**d**) In pattern 2, the follicles are depleted, and regressive follicles remain (HE × 100). (**e**) In pattern 3, the lymph node architecture is totally or mostly effaced (HE × 100).

**Figure 3 diagnostics-12-02001-f003:**
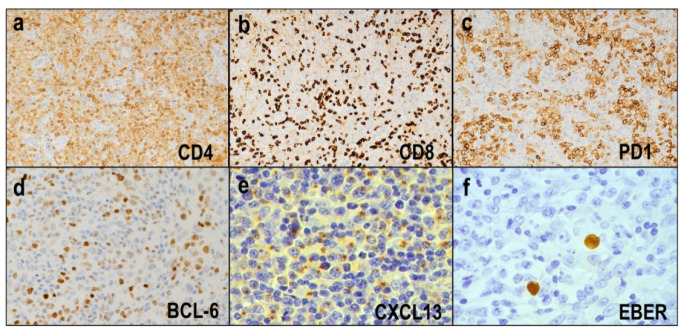
Immunohistochemistry of angioimmunoblastic T-cell lymphoma. (**a**,**b**) Most cases are positive for CD4 (×200) and negative for CD8 (×200). (**c**–**e**) Tumor cells show positive staining for several TFH markers (HE × 400). (**f**) EBV^+^ B cells are present in the majority of cases (×400).

**Figure 4 diagnostics-12-02001-f004:**
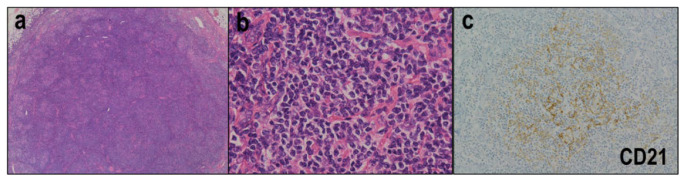
Follicular T-cell lymphoma. (**a**) Lymph nodes show a nodular/follicular growth pattern (HE × 40). (**b**) Aggregations of intermediate-sized neoplastic cells with round nuclei and pale cytoplasm are recognized inside the follicles (HE × 400). (**c**) Immunostaining with CD21 highlights the FDC meshworks in nodular/follicular structures (HE × 200).

**Figure 5 diagnostics-12-02001-f005:**
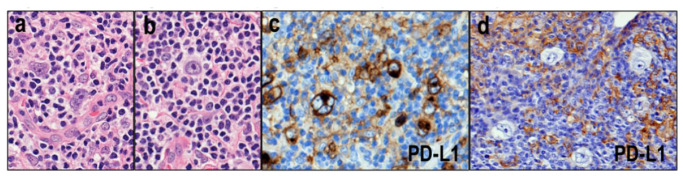
Follicular helper T-cell (TFH) lymphomas with Hodgkin-Reed-Sternberg (HRS)-like cells. (**a**,**b**) The immunoblasts in the background of TFH lymphomas occasionally mimic HRS cells (HE × 400). (**c**) HRS cells show highly frequent PD-L1 expression (×400). (**d**) PD-L1 expression is rare in HRS-like cells (×400).

**Figure 6 diagnostics-12-02001-f006:**
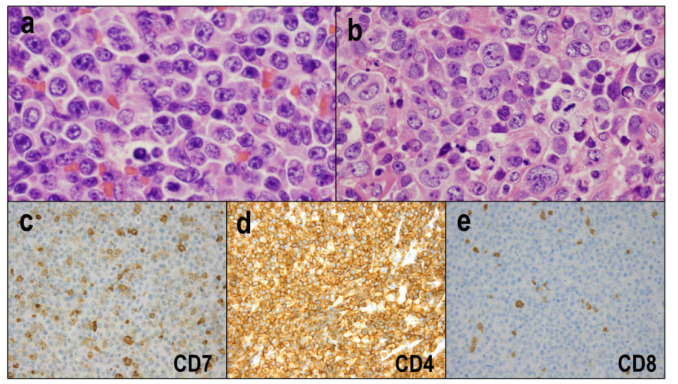
Adult T-cell leukemia/lymphoma. (**a**) Tumor cells are usually medium-to-large and pleomorphic (HE × 400). (**b**) Anaplastic-like cells are occasionally present (HE × 400). (**c**) Tumor cells usually lack CD7 expression (×200). (**d**,**e**) Most cases are positive for CD4 (×200) and negative for CD8 (×200).

**Figure 7 diagnostics-12-02001-f007:**
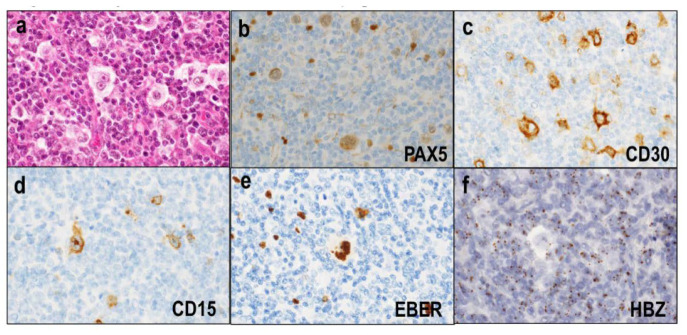
Hodgkin-like adult T-cell leukemia/lymphoma. (**a**) Small-to-medium-sized lymphocytes with mild nuclear atypia diffusely proliferate with scattered HRS-like cells (HE × 400). (**b**) The HRS-like cells are of the B-cell lineage (PAX5 × 400). (**c**–**e**) The HRS-like cells show positive staining for CD30 (×400) and/or CD15 (×400) and are frequently infected by EBV (×400). (**f**) HRS-like cells are not infected with HTLV-1 (×400).

**Figure 8 diagnostics-12-02001-f008:**
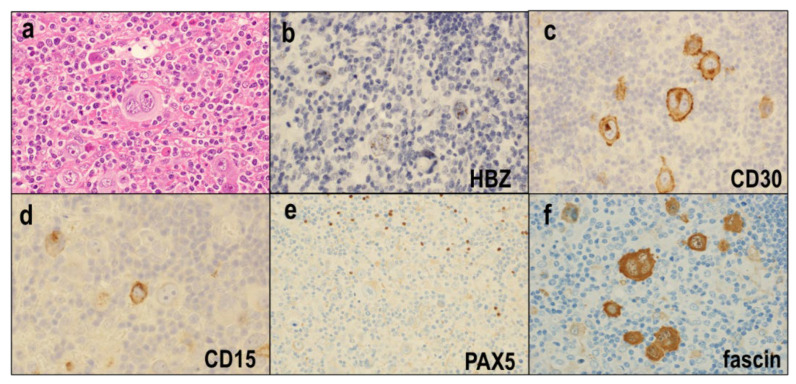
Adult T-cell leukemia/lymphoma with HTLV-1-infected HRS-like cells (ATLL-HH). (**a**) Scattered HRS-like cells are intermingled with abundant inflammatory cells (×400). (**b**) The HRS-like cells of ATLL-HH are infected by HTLV-1. (**c**–**e**) The HRS-like cells show positive staining for CD30 (×400) and CD15 (×400) and are negative for B-cell markers (PAX5 × 400). (**f**) The HRS-like cells are frequently positive for fascin (×400).

**Figure 9 diagnostics-12-02001-f009:**
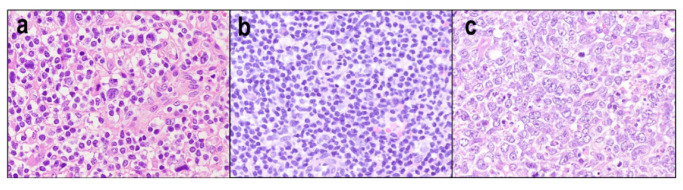
Cytological spectrum of peripheral T-cell lymphoma, NOS. (**a**) Pleomorphic medium- and large-sized type (HE × 400). (**b**) Pleomorphic small cell type (HE × 400). (**c**) Large cell immunoblastic type (HE × 400).

**Figure 10 diagnostics-12-02001-f010:**
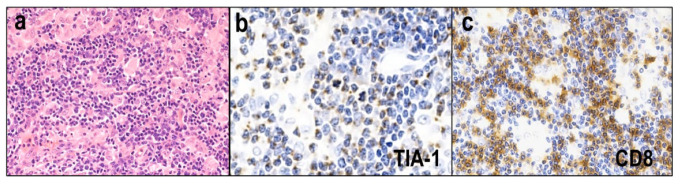
Lymphoepithelioid variant of peripheral T-cell lymphoma, NOS (Lennert lymphoma). (**a**) Numerous clusters of epithelioid histiocytes, and the infiltration of small-to-medium-sized lymphocytes with slight nuclear irregularities (HE × 400). (**b**,**c**) Tumor cells are usually positive for cytotoxic molecule (TIA-1 × 400) and (CD8 × 400).

**Table 1 diagnostics-12-02001-t001:** Translocations and ALK staining pattern of ALK^+^ ALCL.

Translocation	Partner Gene	ALK Staining Pattern	Frequency
t(2;5)(p23;q35)	*NPM1*	Cytoplasmic and nuclear	80%
t(1;2)(q25;p23)	*TPM3*	Cytoplasmic with peripheral intensification	10–15%
inv(2)(p23q35)	*ATIC*	Cytoplasmic	1%
t(2;3)(p23;q12.2)	*TFG*	Cytoplasmic	<1%
t(2;17)(p23;q23)	*CLTC*	Cytoplasmic	<1%
t(2;22)(p23;q11.2)	*MYH9*	Cytoplasmic	<1%
t(2;17)(p23;q23)	*RNF213*	Cytoplasmic	<1%
t(X;2)(q11-12;p23)	*MSN*	Membrane	<1%
t(2;9)(p23;p13.1)	*TPM4*	Cytoplasmic	<1%
t(2;9)(p23;q33)	*TRAF1*	Cytoplasmic	<1%
t(2;11)(p23;qR3)	*EEFIG*	Cytoplasmic	<1%
t(2;8)(p23;q22)	*PABCP1*	Cytoplasmic	<1%

**Table 2 diagnostics-12-02001-t002:** Recurrent genetic alterations and key biomarkers of ALCL.

Genetic Alterations and Biomarkers	Subtype	Notes
CD30 expression	ALCL	Expression in 100% (by definition); target of BV
*ALK* rearrangement	ALK^+^ ALCL	Translocations in 100% (by definition); sensitive to ALK inhibitors
*DUSP22* rearrangement [21]	ALK^−^ ALCL	Translocation in 30%; favorable prognosis
*TP63* rearrangement [21]	ALK^−^ ALCL	Translocation in 8%; dismal prognosis
*JAK1* and *STAT3* [29]	ALK^−^ ALCL	JAK1 mutations in 15% and *STAT3* mutations in ~10%; activation of the JAK-STAT3 pathway

ALCL, anaplastic large cell lymphoma; BV, brentuximab vedotin.

**Table 3 diagnostics-12-02001-t003:** Recurrent genetic alterations and key biomarkers of nodal TFH lymphomas.

Genetic Alterations and Biomarkers	Subtype	Notes
*TET2* mutation [43,44,45,62]	TFH lymphoma	Loss of function mutations in 40–80% of AITL; slightly more frequent in FTCL and nPTCL-TFH
*DNMT3A* mutation [43,44,45,62]	TFH lymphoma	Loss of function mutations in 30% of AITL
*IDH2^R172^* mutation [46,47,62]	AITL and FTCL	Mutations are restricted to AITL (30%) and FTCL; increased DNA methylation
*RHOA^G17V^* mutation [44,48,49,50,62,64]	TFH lymphoma	Mutations in 50–71% of AITL; loss of RHOA GTPase activity
*ITK-SYK* fusion [65]	FTCL	Fusions in 20%; specific for FTCL; sensitivity to the SYC inhibitor

AITL, angioimmunoblastic T-cell lymphoma; FTCL, follicular T-cell lymphoma; nPTCL-TFH, nodal peripheral T-cell lymphoma with TFH phenotype; TFH, follicular helper T cell.

**Table 4 diagnostics-12-02001-t004:** Recurrent genetic alterations and key biomarkers of ATLLs.

Genetic Alterations and Biomarkers	Subtype	Notes
Genomic alterations affecting the TCR/NF-κβ pathway [85,86,87,88]	ATLL	Genetic alteration in up to 90%; genetic alterations of a proximal component of TCR signaling (e.g., *FYN, PLCG1*, and *VAV1*), of the NF-κβ pathway (e.g., *PRKCB* and *CARD11*), and of downstream signaling (e.g., *IRF4* and *RHOA*).
Fusions related to CD28 [89]	ATLL	In-frame fusions in 10%; *CTLA4-CD28* and *ICOS-CD28*; contribute to NF-κβ pathway activation
*PD-L1* amplification [85]	ATLL	Copy number gains in 15%; potential target of the immune checkpoint inhibitor; worse prognosis
*PD-L1* disruption of the 3′-UTR [90]	ATLL	Disruption of the 3’-UTR in 27%; potential target of the immune checkpoint inhibitor
PD-L1 expression [92]	ATLL	Expression in 7.4%; worse prognosis
*CCR4* mutation [85,94,95]	ATLL	Mutations in 30%; up-regulation of CCR4 expression
CCR4 expression [79]	ATLL	Expression in 90%; worse prognosis; good candidate for the anti-CCR4 antibody (mogamulizumab)

ATLL, adult T-cell leukemia/lymphoma; TCR, T-cell receptor; UTR, untranslated region.

**Table 5 diagnostics-12-02001-t005:** Recurrent genetic alterations and key biomarkers of PTCL-NOSs.

Genetic Alterations and Biomarkers	Subtype	Notes
GATA3 and/or CCR expression [111]	GATA3 subgroup	Biological subgroup of PTCL-NOS; 30–40% of PTCL-NOS; worse prognosis
TBX21/CXCR3 expression [111]	TBX21 subgroup	Biological subgroup of PTCL-NOS; 50–60% of PTCL-NOS; better prognosis than that of the GATA3 subgroup
*TP53* mutation and deletion [110,112]	PTCL-NOS	Frequent in the GATA-3 subgroup; worse prognosis
*CDKN2A* deletion [110,112,113]	PTCL-NOS	Frequent in the GATA-3 subgroup; worse prognosis
PTEN deletion [110,112,113]	PTCL-NOS	Frequent in the GATA-3 subgroup; worse prognosis
*FAT1* mutation [114]	PTCL-NOS	Mutations in 39%; worse prognosis
CM expression [100,103,104]	nPTCL-NOS	Expression in 15–40%; aggressive clinical behavior; worse prognosis

CM, cytotoxic molecule; nPTCL-NOS, nodal peripheral T-cell lymphoma, not otherwise specified.

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
