# Peer review of "Pathological and Molecular Features of Nodal Peripheral T-Cell Lymphomas"

_diagnostics, 2022, doi:10.3390/diagnostics12082001_

Round 1

Reviewer 1 Report

Perfect English, concise text describing updated scientific pathology, immunohistochemistry, molecular, and morphological data. 

Author Response

Reply to reviewer 1

We wish to thank the reviewer for her/his comments.

  1. The reviewer gave us a comment that the manuscript is fine.

→ Thank you very much for your favorable comment. It is honor for us to receive such a wonderful comment.

Reviewer 2 Report

Reviewer’s comments to the author:

1. This manuscript only partially summarizes and updates the pathological properties and molecular features of nodal PTCLs. It is suggested that clinical, histopathological, immunophenotypic, genetic, and molecular diagnostic criteria are used to classify these neoplasms for further description. In particular, the differential diagnosis, prognostic factors, and therapeutic considerations of various types of nodal PTCLs are further analyzed.

2. Please describe in detail what is the node PTCL fact in this article review. Should it be emphasized and highlighted what are the continued education and learning objectives and clinical and epidemiological aspects of this manuscript?

3. This article reviews the evaluation and treatment of these common subtypes of nodal PTCL and highlights the summary and update of the pathological and molecular features of nodal PTCLs. How future Intensification of treatment using new drugs or specific monoclonal antibodies may change this reality in the near future. How do enhance the prognostic factors and therapeutic considerations of healthcare team outcomes for reference and R&D goals?

Author Response

Reply to reviewer 2

We wish to thank the reviewer for her/his comments.

  1. The reviewer pointed out that this manuscript only partially summarizes and updates the pathological properties and molecular features of nodal PTCLs. Particularly, the reviewer mentioned that clinical matters (e.g. prognostic factors, therapeutic considerations etc.) have to be further mentioned.

→ Thank you very much for your great opinion. I added Table 2-5 which summarize the key genetic features and biomarkers of each entity. In the tables, prognostic factor and therapeutic options are briefly mentioned.

  1. The reviewer suggested that aim of this manuscript need to be emphasized and highlighted more.

→ Thank you very much for your suggestion. I am afraid the aim of the paper is mentioned in the abstract and conclusion; this paper aims to provide a summary and update of the pathological and molecular features of nodal PTCLs, which will be helpful for diagnostic practice. Perhaps I do not correctly understand your comments. If so, I am deeply sorry for that.

  1. The reviewer suggested that how future intensification of treatment using new drugs or specific monoclonal antibodies may change this reality in the near future and how the prognostic factors and therapeutic considerations of healthcare team enhance the outcomes for reference and R&D goals.

→ Thank you very much for your suggestion. This paper is mainly aimed to summarize and update the pathological and molecular features of nodal PTCLs. Therefore, clinical and therapeutic issues are not described a lot. According to the comment, I added Table 2-5 which briefly mention prognostic factor and therapeutic options.

Reviewer 3 Report

This is a nice review about Peripheral T-Cell Lymphomas (PTCL). The manuscript is well written and logically organized with representative histological images for each subtype and information on their genetic alterations. A summary table for all subtypes regarding biomarkers, gene mutations in oncogenic signaling pathways and epigenetic alterations may be considered. 

Author Response

Reply to reviewer 3

We wish to thank the reviewer for her/his comments.

  1. The reviewer suggested to add a summary table for all subtypes regarding biomarkers, gene mutations in oncogenic signaling pathways and epigenetic alterations.

→ Thank you very much for your suggestion. Accordingly, we added Table 2-5 which summarize the key genetic features and biomarkers of each entity.